# In Situ Aqueous Spice Extract-Based Antifungal Lock Strategy for Salvage of Foley’s Catheter Biofouled with *Candida albicans* Biofilm Gel

**DOI:** 10.3390/gels11010023

**Published:** 2025-01-01

**Authors:** Bindu Sadanandan, Vaniyamparambath Vijayalakshmi, Kalidas Shetty, Adithya Rathish, Harshala Shivkumar, Malavika Gundreddy, Nikhil Kumar Kagganti Narendra, Nethra Machamada Devaiah

**Affiliations:** 1Department of Biotechnology, M S Ramaiah Institute of Technology, Bengaluru 560054, Karnataka, India; v.viju114@gmail.com (V.V.); adithyarathish.11@gmail.com (A.R.); harshalashivkumar@gmail.com (H.S.); malavikagundreddy@gmail.com (M.G.); nikkk26@outlook.com (N.K.K.N.); nethramd2310@gmail.com (N.M.D.); 2Department of Plant Sciences, North Dakota State University, Fargo, ND 58105, USA; kalidas.shetty@ndsu.edu

**Keywords:** antifungal lock therapy, in situ, Foley’s catheter, gel, biofilm, spice extract, *Candida albicans*

## Abstract

*Candida* forms a gel-like biofilm in the Foley’s catheter (FC) causing tenacious biofouling and severe urinary tract infections (UTIs). For the first time, a spice extract-based antifungal lock therapy (ALT) has been developed to inhibit the *Candida albicans* gel matrix in FC. Aqueous extracts of garlic, clove, and Indian gooseberry were used as ALT lock solutions and tested against biofilm-forming multidrug-resistant clinical isolates of *C. albicans*. Reduction in the gel matrices formation in the catheter was confirmed by Point inoculation, MTT assay, CFU, and SEM analysis at 12 and 24 h of incubation. Garlic was effective in controlling both *C. albicans* M207 and *C. albicans* S470; however, clove and gooseberry effectively controlled the latter. As evidenced by CFU assay, there were 82.85% and 99.68% reductions in the growth of *C. albicans* M207 and S470, respectively, at 24 h of incubation. SEM revealed a switch from the biofilm to the yeast mode and a drastic reduction in cell numbers, with mostly clumped or lysed cells. The study will provide an impetus to the development of novel spice extract-based ALT, reducing the selection pressure on the pathogen and lowering antimicrobial resistance. Further research in this area has the potential to leverage clinical applications.

## 1. Introduction

The Center for Disease Control and Prevention categorizes major Hospital Acquired Infections (HAIs) as Central Line-Associated Bloodstream Infections (CLABSIs), Catheter-Associated Urinary Tract Infections (CAUTIs), Ventilator Associated Pneumonia (VAP), and Surgical Site Infections (SSIs) [1]. About 3.2 million people are affected by HAIs in a year, implying that 1 in 31 hospital patients have one of the above-mentioned HAIs. The percentages of hospitalized patients affected by HAIs are 4% in Europe, 2.5–14.8% in Africa, and 10–30% in India. However, the incidence rate for HAIs in India ranges from 11–83%. CAUTIs occur in individuals whose urinary bladder is catheterized or will be catheterized within 48 h and can cause secondary bloodstream infections. They represent about 12–50% of overall hospitalized infections [2]. CAUTIs contribute to almost 80% of nosocomial Urinary Tract Infections (UTIs) [3]. This may be due to the fact that the catheterized patient’s urinary bladder is not emptied, which serves as a nidus for infection. CAUTIs can be caused by bacteria, such as *Escherichia coli*, *Klebsiella*, *Enterococcus*, *Pseudomonas aeruginosa*, *Enterobacter* spp., *Staphylococcus* spp., *Proteus mirabilis*, or fungi—mostly *Candida albicans*, *Candida glabrata*, *Candida tropicalis*, and rarely, *Cryptococcus neoformans* [4], which either enter through the bloodstream or retrograde via the urethra or bladder. Though the major causative agent of CAUTIs is a bacterial infection, fungi also contribute significantly, especially in hospital settings where there are patients already suffering Candidemia or who are immunocompromised. Amongst fungi, *Candida* (23%) species are the most common pathogen, with such infection being attributed to almost one quarter of disease cases [5,6]. These organisms form a self-produced gel matrix composed of polysaccharides, proteins, and DNA, called biofilms. They are the fourth leading hospital-associated infection. The highest rate of morbidity and mortality in CAUTIs is associated with *Candida* species [7,8], with more in the ICU setting [9,10].

*Candida* species in UTIs have an overall prevalence rate of 11.2%, with a high incidence rate of 59.7% in the age group between 21 and 30 years [11]. Pregnant women have a higher rate of UTIs (33%) than married women (27%) or single women (15%). Married males exhibited a higher (18%) prevalence of UTIs compared to single men (7%) [12]. *Candida*-associated CAUTIs become more serious in patients when they are drug-resistant. Multi-drug *Candida* species are more prevalent in hospital infections. People with UTIs caused by multidrug-resistant *Candida* species will have to try multiple drugs, which can lead to tissue and systemic toxicity, ultimately resulting in complicated side-effects.

*Candida* biofilm is a gel-like extracellular polysaccharide (EPS) matrix that anchors to the surface of medical devices. It can facilitate adhesion to medical devices causing their fouling and rendering them unfit for use. EPS acts as a natural gel that binds the microbial cells together to form cell aggregates or biofilm to protect the microorganism against abiotic and biotic stress. The formation of biofilm is most common in urinary catheters, especially FC [13], which may lead to severe cervical canal infections [14], urinary tract infections [15], and act as a cause for high-risk nosocomial candiduria [16]. These biofilms are made up of microbial cells along with gel-like EPS, which can vary in thickness and percentage composition. Infections caused by *Candida* species in the catheter lead to the removal of the infected catheter followed by antifungal therapy and then replacement with a new one [17]; however in some patients, it is not advisable for the catheter to be removed.

The lab on-a-chip technique has been used in recent times to study different aspects of *Candida* biofilms. Couttenier et al. studied the bending stiffness of *C. albicans* hyphae. They developed a microfluidic device to facilitate this. The cell morphology of *C. albicans* is defined by its mechanical properties and the bending stiffness varies depending on the environmental stress or genetic modifications. Their results proved that the cell wall of *C. albicans* is highly elastic and the organism has the ability to remodel entire hyphae in a matter of minutes [18]. In another study, a high-throughput microfluidic chip was used to screen 50,520 small molecules in combination with amphotericin B. Of the compounds screened, 10 showed a 30% improvement in the antifungal effect [19]. An in vitro microphysiological intestine-on-a-chip model was developed to study the microbial interaction of *Lactobacillus rhamnosus* and *C. albicans.* The colonization of *L. rhamnosus* on the chip reduced *C. albicans*-induced tissue damage and lowered the fungal load. This model facilitated the immune characterization and study of *Candida* pathogenesis [20]. Claudel et al. developed a lab-on-a-chip bio-impedance cytometry model to detect the electrical signature of a single yeast cell (*Saccharomyces cerevisiae*). This method was able to detect single cells and measure their size as well as their electrical properties [21].

ALT is used to clear the catheters of biofilms by applying a continuous infusion of antimicrobials, called ALT lock solutions, at high concentrations into the catheter lumen for extended periods [22]. To salvage the catheters, ALT can be applied either ex situ/ex vivo or in situ/in vivo. Some of the conventional ALT lock solutions for fungal biofouling are fluconazole [23], caspofungin [24,25], micafungin [24,26], amphotericin B [25], ethanol [27], and silver nanoparticles [28]. The concentration of conventional lock solutions is 10x times more than that of the therapeutic range. The use of high doses of conventional antimicrobials in the in vivo ALT has severe side-effects on the patient. It is also not very effective against multidrug-resistant organisms like *C. albicans*. This calls for an alternative ALT lock solution that can prevent toxic side-effects.

Spice, condiment, and herb extracts of garlic [29,30,31], cinnamon, clove, jasmine, rosemary [32], and Indian gooseberry (amla) extracts have potential antifungal effects with no known toxicity [33]. Therefore, antifungal spice extracts have prospects as natural and ALT lock solutions in preventing biofouling of catheters and thereby mitigating fungemia and septicaemia in affected patients. However, the effectiveness of ALT would depend on the choice, concentration, and time of incubation of the antifungal agents.

High-biofilm-forming drug-resistant clinical isolates of *C. albicans* cause extensive biofouling of medical devices and are a challenge to control using conventional antimycotic agents. The objective of this study was to address this issue by using three spice extracts—garlic (*Allium sativum*), clove (*Syzygium aromaticum*), and Indian gooseberry (*Phyllanthus emblica*)—as ALT lock solutions in the development of an in situ ALT to control the biofouling of FC by high-biofilm-forming multidrug-resistant clinical isolates *C. albicans* M207 and S470. Garlic, clove, and gooseberry have multiple antimicrobial properties. Garlic exhibits antifungal, antibacterial, anti-oxidant, anti-cancer, and anti-inflammatory properties [31,34]. Garlic has demonstrated antimicrobial activity against the ESKAPE group of bacterial pathogens [35,36]. Garlic also exhibits antifungal activity against *C. albicans*, *Cryptococcus neoformans*, *Ascosphaera apis*, *Aspergillus niger*, *Fusarium oxysporum*, *Torulopsis*, *Trichosporon*, and *Rhodotorula* spp. [37,38]. Allicin is the main component in garlic, constituting 70–80% of sulfur compounds. It has antimicrobial potential against bacteria, fungi, viruses, and parasites. This is due to its reaction with the thiol group of enzymes which can affect essential metabolism. The disruption of garlic releases enzymes like alliinase that convert alliin into allicin. However, allicin is highly unstable and can decompose into other stable organosulphur compounds, including allyl sulfide, diallyl sulfide, triallyl sulfide, ajoenes, etc, which also have a similar effect to allicin [39]. Garlic also contains free amino acids (1.2%), fiber (1.5%), protein (2%), organosulphur compounds (2.3%), carbohydrates (28%), and water (65%) [40]. Clove is known for its antifungal, antibacterial, antioxidant, analgesic, anti-infective, and anti-inflammatory properties [41,42,43]. It is effective against food-borne pathogens, such as *Staphylococcus*, *Escherichia*, *Listeria*, *and Salmonella*. It is also effective against *Pseudomonas*, *Candida*, *and Klebsiella* species. Clove has bioactive components, such as eugenol acetate, eugenol, caryophyllene, gallic acid, ellagic acid, biflorin, kaempferol, and quercetin. Gooseberry shows antioxidant, anti-cancer, chemoprotective, anti-viral, immunomodulatory, anti-aging, and anti-inflammatory properties. It is effective against *Staphylococcus*, and other multidrug-resistant cultures, due to the high level of phenolics. The phytochemicals present in gooseberry are gallic acid, ellagic acid, emblicanine A, emblicanine B, phyllantine, phyllatidine, quercetin, etc. [44]. The antimycotic activity of these extracts on *C. albicans* M207 and S470 was studied by point inoculation, MTT assay, colony forming unit assay, and SEM.

Previous studies with ALT have used conventional antimycotic agents. Studies with plant extracts have not been extended to ALT. This is the first study of its kind to investigate spice extract-based ALT developed to control the growth of biofilm-forming multidrug-resistant clinical isolates of *C. albicans* in Foley’s catheter. Further in situ studies on aqueous spice extract-based ALT would pave the way for its implementation in clinical practice.

## 2. Results and Discussion

### 2.1. Foley’s Catheter

Urinary catheters have been an important part of medical care since the invention of the FC. A typical FC has a capacity of 10 mL with a length of 400 mm. It has two channels, one called the drainage channel to discard the urine and a second inflation channel which helps to retain the catheter in the bladder [45]. The benefits of the FC also come with its innate risks. When used frequently and for prolonged periods, it can become a niche for various microorganisms, especially biofilm-formers that can become a lasting source of acute to severe Urinary Tract Infections (UTIs) and other life-threatening infections.

### 2.2. Point Inoculation

Large spreading biofilm phenotypes of *C. albicans* M207 and S470 cultures were grown in 12 and 24 h control plates with the catheter section and, as expected, more growth was observed at 24 h. Both the center and peripheral sections of the catheter when point-inoculated on TSA media exhibited a similar extent of growth, indicating uniform growth of the culture throughout the catheter.

Point inoculation of the garlic extract-treated section of the catheter with *C. albicans* M207 showed substantial inhibition in growth along with a drastic regression of the biofilm for both 12 and 24 h of incubation (Figure 1).

Catheters with *C. albicans* S470 were treated with extracts of garlic, clove, and gooseberry extracts. All three extracts were found to be effective in controlling the biofilm at 12 and 24 h of incubation (Figure 1). However, among them, garlic was the most effective, followed by clove and then gooseberry.

### 2.3. MTT Assay

MTT assay was also performed for the center and periphery sections of the catheters of the 12 (Figure 2, Panel 1) and 24 h grown cultures (Figure 2, Panel 2) for the reconfirmation of uniform growth in the catheter. This assay also helps to directly confirm the viability and indirectly the inhibition of *C. albicans* M207 against garlic extract and of *C. albicans* S470 against garlic, clove, and gooseberry extracts. Amongst these three extracts, for *C. albicans*, S470 the most effective was garlic, followed by clove and gooseberry.

For 12 h of incubation, *C. albicans* M207 showed 86% viability at the center and 89% at the periphery region when treated with garlic. However, at 24 h of incubation, on treatment, the viability decreased to 71% at the center and 72% at the periphery regions. *C. albicans* S470 at 12 h of incubation with the extracts showed a percentage viability of 75% for garlic, 92% for clove, and 96% for gooseberry at the center, and a percentage viability of 76% for garlic, 86% for clove, and 94% for gooseberry at the periphery region. However, at 24 h of incubation, the percentage viability was only 45% for garlic, 69% for clove, and 62% for gooseberry at the center, and at the periphery, 48% for garlic, 57% for clove, and 62% for gooseberry. The targeted extracts were found to be efficacious in inhibiting the biofilm of both the cultures in the center as well as the peripheral regions of the catheter with 24 h application being more effective than 12 h.

The MTT assay confirmed the antimicrobial activity of the spice extracts against the tested multidrug-resistant clinical isolates of *C. albicans*.

### 2.4. Colony-Forming Units (CFUs)

The quantity of CFUs was determined for *C. albicans* M207 with garlic extract (Figure 3: Panel 1, Table 1) and for *C. albicans* S470 (Figure 3: Panel 2, Table 2) with garlic, clove, and gooseberry at 12 and 24 h of incubation.

Garlic was very effective in inhibiting *C. albicans* M207 and S470, followed by clove and gooseberry for the latter.

It can be observed that 12 and 24 h of incubation for *C. albicans* M207 with garlic extract had the same % kill of 82.35% at 10^−1^ dilution. At the same dilution, for *C. albicans* S470, inhibition with garlic was 98.42% and 99.68% for 12 and 24 h, respectively. The percentage kill for clove was 97.6% and 95.54% for 12 and 24 h, respectively, and the percentage kill for gooseberry was 97.6% and 95.6% for 12 and 24 h, respectively.

The colonies appeared to be smaller and more in number at 12 h of incubation; however, at 24 h the colonies appeared to be bigger, and less in number. The higher dilutions of the CFU (10^−2^, 10^−3^, 10^−4^) are shown in Appendix A.

### 2.5. Scanning Electron Microscopy

Since the surface area exposed is less in the cross-section (Appendix A) of the catheter as opposed to the longitudinal section, the details revealed in the SEM images of the latter were far superior to the former, especially with respect to the morphology of the cells and the EPS gel. Abundant cells embedded in the thick EPS gel matrix can be observed in the control images of *C. albicans* M207 and S470 at 12 and 24 h (Figure 4) of incubation. In the images of *C. albicans* M207 treated with garlic extract for 12 h, we can observe clumped and dead cells enmeshed in the almost negligible gel matrix; however, at 24 h of incubation, the patches of clumped cells have mostly reduced, revealing the distorted EPS gel in the background. Likewise, the micrographs of the garlic, clove, and gooseberry-treated *C. albicans* S470 for both time durations also revealed clumped and dead cells in the diminished EPS gel. Here too, garlic was the most effective in inhibiting the pathogenic yeast. In the images of the clove- and gooseberry-treated catheter sections, a few cells embedded in the EPS gel are still seen.

The SEM analysis confirms the effectiveness of the spice extracts against *C. albicans* M207 and S470, which was previously observed through the Point inoculation, CFU, and MTT assays.

### 2.6. Discussion

Biofilm and the associated pathogen infections are a major problem in implants, especially in catheters, and are recalcitrant to conventional antimicrobials [46]. The major classes of conventional antifungals are azole (fluconazole, voriconazole, itraconazole, etc.), echinocandin (caspofungin, micafungin, anidulafungin, etc.), polyene (amphoterecin B, nystatin, natamycin, etc.), allylamine (terbinafine, naftifine, tonaftate, etc.), and miscellaneous (griseofulvin, flucytosine, etc.). Conventionally, to salvage the catheter in patients, antimycotics such as caspofungin and amphotericin B have been used in antimicrobial lock therapy [47]. For lock therapy, a lock solution should have the following characteristics: high stability, low potential to resistance, cost-effective, non-toxic, ability to penetrate the EPS gel, target specificity [27]. Conventional antimicrobials such as Caspofungin and Amphotericin B have been used previously as lock solutions in inhibiting *Candida* organisms [48]. Azoles such as fluconazole and voriconazole are effective in removing *Candida* biofilms [48]. Nikkomycin Z in combination with echinocandins is reported as a good possible adjuvant in lock therapy [49]. However, in another study with the same agents, the results were found to be negative [50]. Some of the classes of antifungal agents and their effects on *Candida* are as follows: Azoles can inhibit ergosterol synthesis by blocking lanosterol 14α demethylase; echinocandins affect the cell wall by blocking β-glucan synthesis; polyenes attack the plasma membrane binding to ergosterol; allylamine inhibits squalene epoxidase, which is an essential enzyme in the ergosterol pathway. Griseofulvin, a polyketide miscellaneous class of antifungal agents, blocks mitosis by preventing the synthesis of microtubules and microfilaments. Flucytosine, a nucleotide analogue also classified under miscellaneous antifungals, inhibits the synthesis of nucleic acids (Figure 5).

Repeated use of conventional ALT can be harmful to the body due to the high concentrations of the ALT drugs and their side-effects. Previous publications have shown the effective inhibition of *Candida* species with garlic [51], clove [32], and gooseberry extracts [52] as alternative therapies. The major component in garlic extract is allicin, in gooseberry is gallic acid, and in clove is eugenol or ellagic acid [53]. We have also obtained similar results through LCMS in our previous study. In clove, however, the major compound detected was ellagic acid [51]. The active principles of garlic (allicin), clove (eugenol and ellagic acid), and gooseberry (gallic acid) enter the cells by simple diffusion. Garlic extract damages the cell wall and causes cell collapse. In its multi-mode action, it affects lipid synthesis, reduces oxygen consumption, inhibits succinate dehydrogenase in the Krebs cycle, and inactivates thiol peptide (glutathione) and proteins (glutathione peroxidase, glutathione reductase, coenzyme A) that act as innate antioxidants, leading to oxidative stress, thereby also triggering ROS generation. It can also alter gene expression involved in oxidative reduction processes, damage mitochondria, and downregulate the ECE1 virulence factor that encodes candidalysin [54]. The extract can also affect the sodium-potassium pump and inactivate quorum-sensing genes. The clove extract binds to ergosterol on the cells and disrupts the cell membrane. It enters the cell and inhibits toxin production, releases intracellular components like radicals, cytochrome C, ions, protein, nucleic acid, etc., and affects the transport of ions and ATP, leading to cell death. It is also known to inhibit adhesion and biofilm formation along with inhibiting toxin production [55,56]. The components in gooseberry extract destroy the cell wall, leading to cytoplasm leakage, damaging protein, DNA, and RNA, and disrupting enzymatic activity. Of the components, alkaloids alter the genetic material of microbes, phenols like ellagic acid and gallic acid control the protein and lipid ratio, flavonoids inhibit RNA synthesis, and tannins inhibit oxidative phosphorylation [57,58]. The antifungal activity of these active principles of garlic, gooseberry, and clove depends on the quantity of the major components present; however, it is difficult to identify the specific site of action as several interactive reactions occur simultaneously [55,59,60].

Hence, we used these aqueous extracts as ALT lock solutions in place of first-line antimycotics to inhibit biofilm buildup of multi-drug-resistant clinical isolates of *C. albicans* M207 and S470. CFU and MTT assay were used to monitor the cell viability and metabolic activity of the *Candida* cultures on treatment with the spice extracts [61,62]. In our earlier publication, we observed that *C. albicans* biofilm, when grown on a polystyrene surface, has an MIC 50 of 1 mg at 12 h of incubation for aqueous garlic extract and at 24 h, persister cells further enhanced the growth of *C. albicans*. In the coculture of *C. albicans* M207 and *E. coli* until 12 h, the biofilm was mostly contributed by the latter; however, at 24 h, the former overcame the inhibitory effect of the latter and the biofilm was mostly contributed by *C. albicans* M207 [30]. However, in the present study, when garlic-treated *C. albicans* M207 was grown in the FC, even up to 24 h, a significant reduction in the viability of cells was observed, mostly due to the difference in the gel matrix in the lumen of the silicone elastomer catheters, as opposed to growth on the surface of the silicone elastomer disks. In all probability, the EPS gel in the catheter is still in its exponential phase and has not developed into a fully mature one [63].

As in this study, Mukherjee et al., 2009 in SEM analysis, also observed debris and dead cells on treatment with conventional antimicrobials [64]. Ionesce et al., 2021 observed a uniform growth of biofilm in the intravesical and intraluminal FC; however, cluster formation was observed in the latter [13]. A comparison of our study with published work is presented in Table 3.

The whole spice extracts used in the study showed very effective inhibition of *C. albicans* in the FC. These extracts have multiple active principles present in them which are responsible for their antimicrobial property. However, purified active compounds alone or in combination with first-line antimycotics may play a very important role as future therapeutics for the treatment of *Candida*-infected catheters in vivo.

## 3. Conclusions

The growth of *C. albicans* was induced on FC and effectively controlled using garlic, gooseberry, and clove in an ALT setup. Though ALT has its strengths, the limitations are that the success of the therapy depends on the species and the strain of pathogen involved. In our case, the *C. albicans* M207 strain can only be inhibited with aqueous garlic extract, whereas the *C. albicans* S470 strain can be inhibited with garlic, gooseberry, and clove. Multiple doses may assist in eliminating *C. albicans* species in the catheter. The dose depends on the eukaryotic nature and the ergosterol availability on the species’ cell wall. Since no organic solvents are used in the extract preparation, it is environmentally friendly and incorporates intrinsic safety into the therapeutic preparation. As in precision medicine, it is not just the pathogenic species that needs to be considered but also the strain. Hence, the future scope of the spice extract-based ALT would involve testing with other pathogenic species of yeasts and bacteria. In vivo studies in the future will also help in customizing solutions for the elimination of cells within EPS gels, thereby avoiding catheter-related infections. The ALT method developed is rapid, affordable, easily adaptable, and scalable worldwide across communities. The study offers a viable basis for developing and creating natural in situ ALT treatment options with aqueous garlic extract, either alone or as a robust combinatorial and synergistic design to treat drug-resistant indwelling catheter-associated *Candida* biofilm infections.

## 4. Materials and Methods

### 4.1. Yeast Cultures

Clinical isolates of *C. albicans* M207 and S470 were used in the study. *C. albicans* M207 was isolated from the umbilical vein catheter of a female baby, and *C. albicans* S470 was isolated from the sputum of a female patient. Both the patients were affected with invasive Candidiasis. The cultures were kindly provided by the Microbiology Laboratory, M S Ramaiah Medical College and Teaching Hospital, Bengaluru, Karnataka, India.

Ethical clearance was not required as no human subjects were involved in the study.

### 4.2. Growth Conditions

*C. albicans* M207 and S470 were sub-cultured on Trypticase Soy Agar (TSA) medium and incubated at 32 °C for 24 h. The clinical isolates were stored as glycerol stocks (15% *v*/*v*) at −20 °C for short-time experiments; however, the mother cultures were stored at −86 °C.

### 4.3. Catheter

A silicone-based FC (RUSCH) was used for the study. The cultures were grown in the lumen of the catheter for the induction of biofilm and to test the antifungal lock therapy.

### 4.4. Extract Preparation

The shelled garlic bulbs, fresh pitted Indian gooseberry fruit (amla), and whole clove buds were carefully chosen for the study as they are known to have antimicrobial properties. These three natural sources were chosen as our previous studies have demonstrated that aqueous garlic extract can effectively control biofilms of *C. albicans* M207 and *C. albicans* S470 and clove and Indian gooseberry can also control the latter [30,51]. Garlic, clove, and Indian gooseberry were locally sourced and authenticated by the Pharmacognosy Department, The Himalaya Drug Company, Makali, Bengaluru. Initially, the samples were washed in tap water followed by sterile water and air-dried before extraction. A weight of 10 g each of shelled garlic cloves and pitted gooseberry fruit was crushed in a pestle and mortar with 5 mL of sterile water, whereas a weight of 5 g of powdered clove was dissolved in 10 mL of sterile water. The extracts were centrifuged at 1000 rpm for 10 min at 4 °C. The supernatant was filtered through Whatman filter paper and was used for further studies [30]. The crude extracts have been characterized and the active principles have been detected via LCMS analysis in our previous study [51].

### 4.5. Pre-Inoculum

A loopful of *C. albicans* M207 and of S470 were inoculated separately in test tubes with 5 mL of TSB medium, sealed, and incubated overnight in a shaker incubator at 32 °C. The next day, the optical density was read, and the culture inoculum was adjusted to a cell count of 1 × 10^6^ cells/mL for further experiments.

### 4.6. In Vitro Induction of Candida albicans Biofilm in Foley’s Catheter and Spice Extract-Based Antifungal Lock Therapy (ALT)

Pre-inoculated cultures were adjusted to a cell density of 1 × 10^6^ cells/mL. The extracts of garlic (200 mg dry weight), clove (43 mg dry weight), and Indian gooseberry (86 mg dry weight), were prepared and kept ready. The in vitro ALT was applied by mixing the culture and the extract in a ratio of 1:1 along with a control plate in parallel. *C. albicans* M207 was treated with garlic extract and *C. albicans* S470 was treated with garlic, gooseberry, and clove extracts, respectively. FC tube was cut into 10 cm long pieces. One end of the catheter was sealed with parafilm, and the culture-spice extract mixture was added into the catheter from the other end and sealed with parafilm. The catheter was then incubated in an incubator at 32 °C for 24 h. The next day, the culture was removed from the catheter and washed with phosphate-buffered saline (PBS). Using a sterile blade, thin sections of the catheter were cut at the center and periphery regions and used for further experimentation.

### 4.7. Point Inoculation

Thin sections of the catheter cut from the center and periphery region of the catheter were placed at the center of TSA plates and incubated for 16 h at 32 °C. The growth of the culture biofilm was observed the following day.

### 4.8. MTT Assay

Thin sections of the center and periphery regions of the catheters with 12 and 24 h grown cultures were placed in a 96-well microtiter plate. MTT (5 mg/mL) solution was added to each of the wells and incubated for 3 h. After the incubation period, the MTT was discarded, and acidified isopropanol was added to each well and incubated for 20 min. The wells were vigorously mixed and thereafter 100 µL aliquot was transferred to a fresh well and the absorbance was read at 540 nm using a microplate reader (BioTek, Bengaluru, India).

### 4.9. Colony-Forming Units (CFUs)

*C. albicans* cultures were grown inside the Foley’s catheter as discussed in Section 4.6. The catheter cross-sections were added into Eppendorf tubes with 1 mL PBS each. With vigorous mixing, the cells from the catheter sections were detached into the PBS solution. A serial 10-fold dilution of the control and treated samples was performed. Each of the dilutions up to 10^−4^ was vortexed thoroughly and a volume of 100 µL of the serially diluted samples was evenly spread on TSA plates by the spread plate method and the plates were incubated at 32 °C for 24 h. After incubation, the number of colonies was counted by a conventional manual method, and the CFU concentration was calculated as per Equation (1). The images of the Petri plates were also captured and documented.
CFU/mL = No. of colonies × Dilution factor/Volume of culture plated(1)

The percentage kill of the *C. albicans* cultures was calculated as mentioned below.
% Kill = (Control-Treated)/Control × 100(2)

### 4.10. Scanning Electron Microscopy

After the incubation periods of 12 and 24 h, the lumen of the catheter was washed with PBS and the catheter was sliced vertically and horizontally to obtain a longitudinal section (LS) and cross-section (CS). The LS and CS were then dipped in a 4% glutaraldehyde fixative for 1 h followed by a PBS wash. The sections were further dehydrated in a series of ethanol washes and later air-dried and stored for further studies [51]. The sections were mounted on an SEM stub and sputtered with gold. The images were captured at 2500× magnification using a JSM-IT300 scanning electron microscope at AFMM, Indian Institute of Science, Bengaluru, India.

### 4.11. Statistical Analysis

Three independent trials were performed for all the experiments. Values of broth microdilution were expressed as mean ± standard deviation. The treated samples were compared with their respective controls. Statistical analysis was conducted by two-way ANOVA with a *p*-value of ≤0.05 as statistically significant.

## Figures and Tables

**Figure 1 gels-11-00023-f001:**
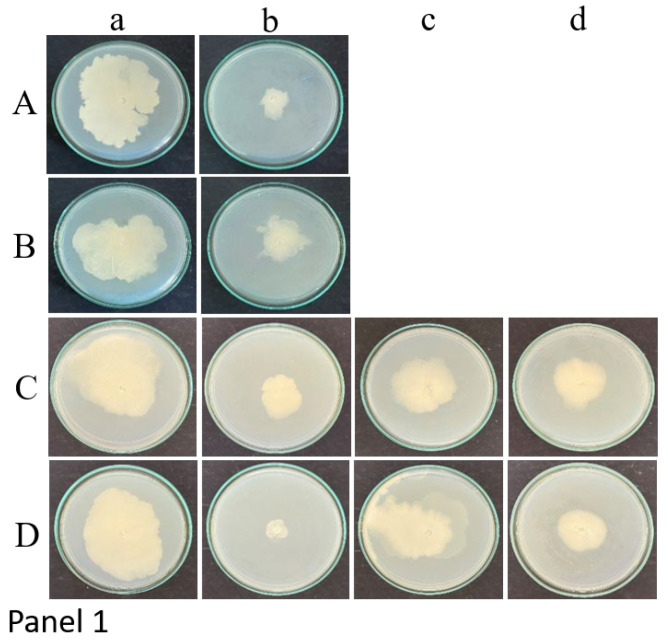
Panel 1: Point inoculation of the catheter sections at 12 h of incubation for (a) control, (b) garlic, (c) clove, and (d) gooseberry extracts. (A) *C. albicans* M207 center, (B) *C. albicans* M207 periphery, (C) *C. albicans* S470 centre, (D) *C. albicans* S470 periphery. Panel 2: Point inoculation of the catheter sections at 24 h of incubation for (a) control, (b) garlic, (c) clove, and (d) gooseberry extracts. (A) *C. albicans* M207 center, (B) *C. albicans* M207 periphery, (C) *C. albicans* S470 centre, (D) *C. albicans* S470 periphery.

**Figure 2 gels-11-00023-f002:**
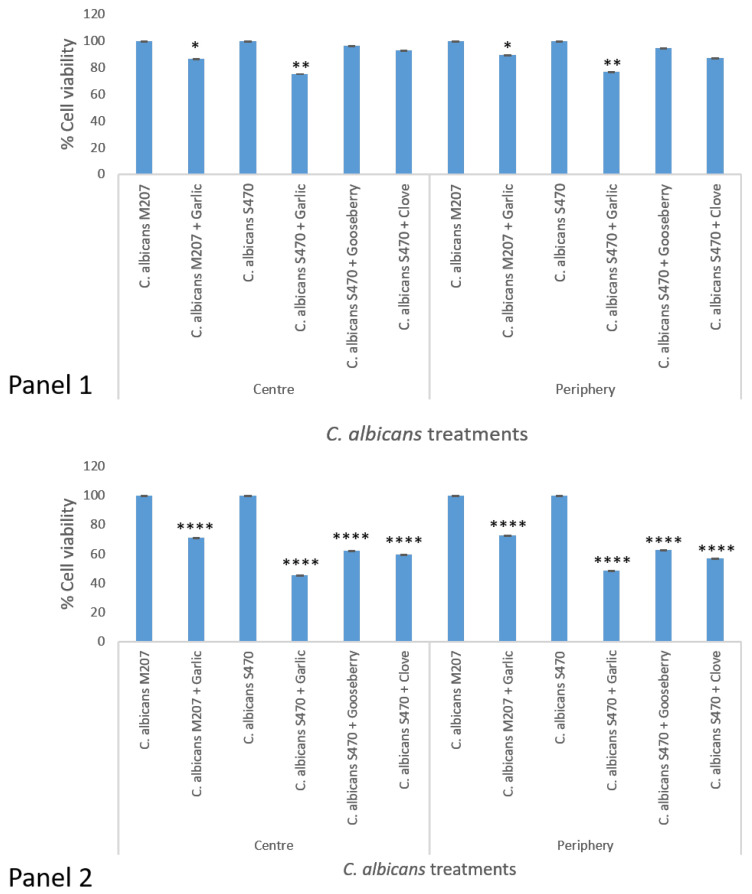
Panel 1: MTT assay of center and periphery regions of the catheter having *C. albicans* M207 and *C. albicans* S470 grown for 12 h and treated with garlic, gooseberry, and clove. * *p* ≤ 0.05, ** *p* ≤ 0.01. The asterisk indicates a significant difference with respect to the control. Panel 2: MTT assay of center and periphery regions of catheter having *C. albicans* M207 and *C. albicans* S470 grown for 24 h and treated with garlic, gooseberry, and clove. **** *p* ≤ 0.0001. The asterisk indicates a significant difference with respect to the control.

**Figure 3 gels-11-00023-f003:**
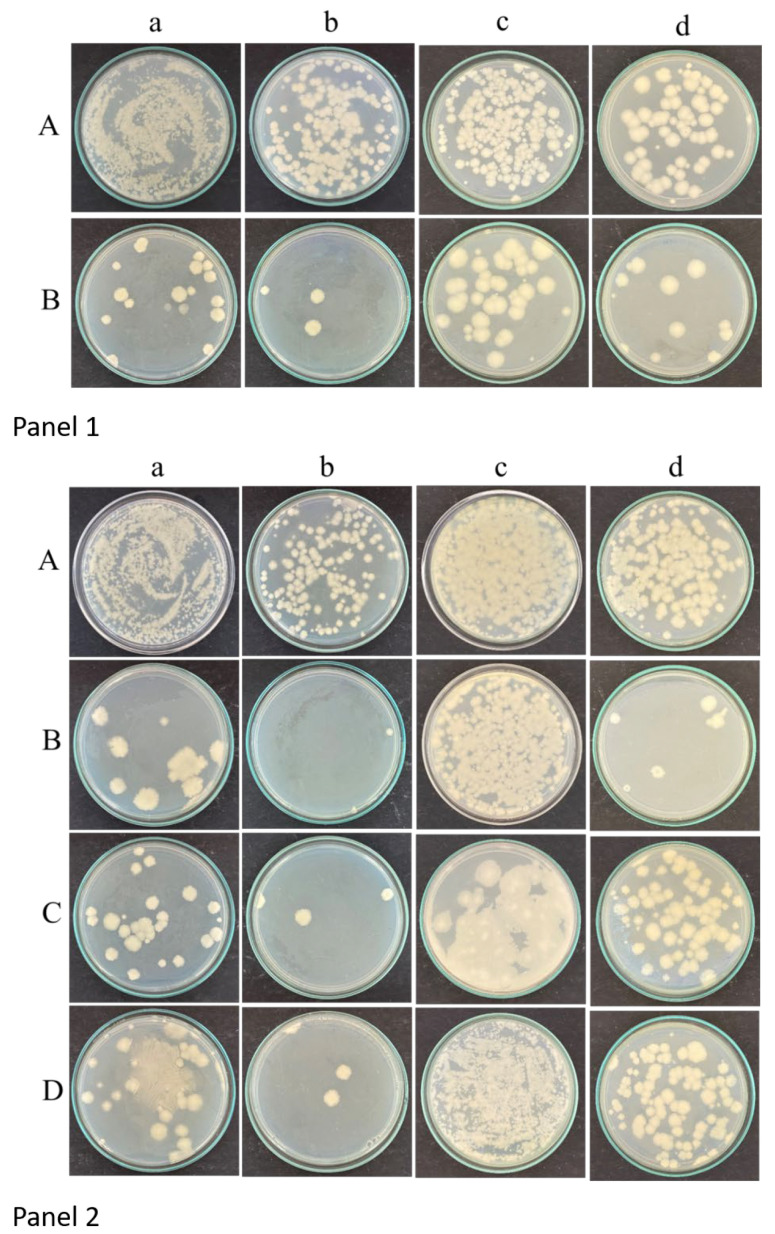
Panel 1: CFUs of (A) *C. albicans* M207 control treated with (B) garlic extract at 12 and 24 h of incubation. (a) 12 h neat, (b) 12 h 10^−1^ dilution, (c) 24 h neat, (d) 24 h 10^−1^ dilution. Panel 2: CFUs of (A) *C. albicans* S470 control treated with (B) garlic, (C) gooseberry, and (D) clove extracts at 12 and 24 h of incubation. (a) 12 h neat, (b) 12 h 10^−1^ dilution, (c) 24 h neat, (d) 24 h 10^−1^ dilution.

**Figure 4 gels-11-00023-f004:**
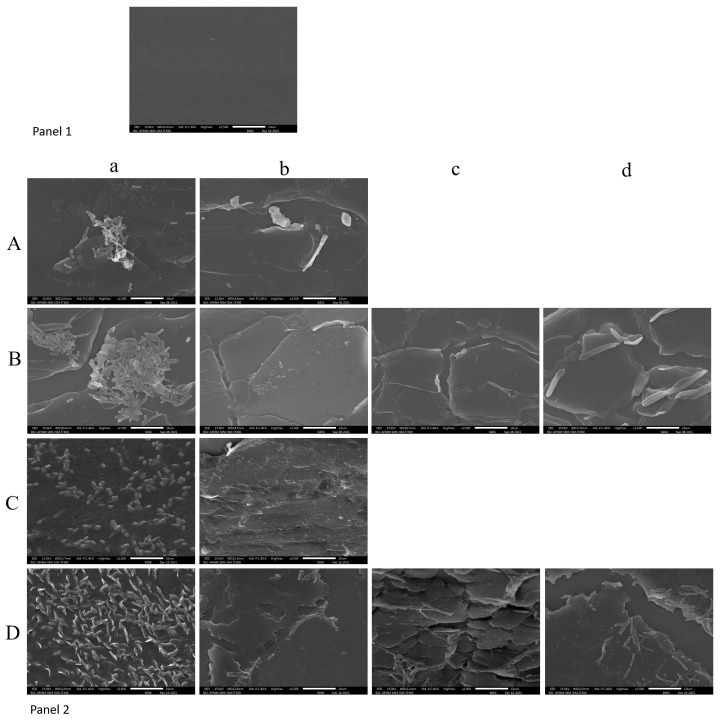
SEM analysis of the longitudinal section of the catheter. Panel 1: Blank catheter. Panel 2: (A) *C. albicans* M207 at 12 h, (B) *C. albicans* S470 at 12 h, (C) *C. albicans* M207 at 24 h and (D) *C. albicans* S470 at 24 h. (a) Control, (b) garlic-treated, (c) gooseberry-treated, (d) clove-treated.

**Figure 5 gels-11-00023-f005:**
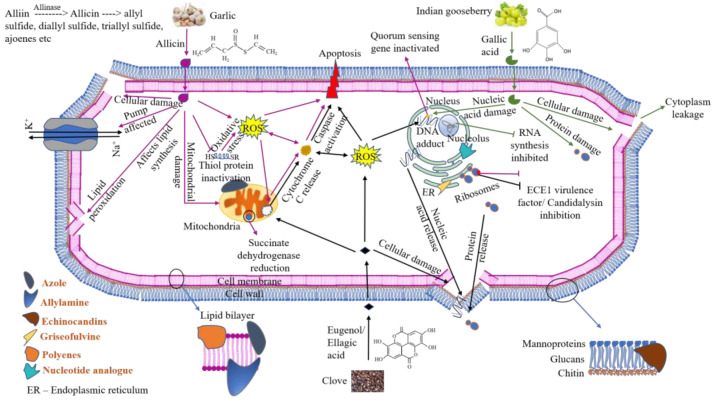
The proposed mechanism of action of allicin, eugenol/ellagic acid, and gallic acid of garlic (pink arrow), clove (black arrow), and gooseberry (green arrow), respectively, on *Candida* sp. Allicin causes cellular damage, affects the sodium–potassium pump, affects lipid synthesis, causes lipid peroxidation, inactivates thiol peptide (glutathione) and proteins (glutathione peroxidase, glutathione reductase, coenzyme A) that act as innate antioxidants, leading to oxidative stress, thereby also triggering ROS generation, damages mitochondria, reduces succinate dehydrogenase, produces ROS, which, in turn, cause cytochrome C release, caspase activation and apoptosis, inhibits ECE1 virulence factor/candidalysin, and inactivates quorum-sensing genes. Eugenol/ellagic acid causes cellular damage, releases cytochrome C, leading to caspase activation and apoptosis, produces ROS, releases nucleic acid and proteins, inhibits ECE1 virulence factor/candidalysin, and forms DNA adducts. Gallic acid causes cellular damage, leading to cytoplasmic leakage, damage to nucleic acid and proteins, and inhibits RNA synthesis. Classes of conventional antimycotics (azole, allylamine, echinocandins, polyenes, the miscellaneous class—griseofulvin, and nucleotide analogues) and their binding sites are also depicted in the figure.

**Table 1 gels-11-00023-t001:** CFU results of *C. albicans* M207 treated with garlic extract at 12 and 24 h of incubation.

Dilutions	*C. albicans* M207—CFU/mL					
	12 h			24 h		
	Control	Garlic Treated	% Kill	Control	Garlic Treated	% Kill
Neat	* Not countable	1510	-	1950	560	71.28
10^−1^	1700	300	82.35	7000	1200	82.85
10^−2^	26,000	0	100	50,000	5000	90
10^−3^	60,000	0	100	220,000	10,000	95.45
10^−4^	200,000	0	100	1,500,000	0	100

* Colonies are too small and too many to be counted.

**Table 2 gels-11-00023-t002:** CFU results of *C. albicans* S470 treated with garlic, clove, and gooseberry extracts at 12 and 24 h of incubation.

Panel 1							
**Dilutions**	**12 h**						
	**Control**	**Garlic Treated**	**% Kill**	**Clove Treated**	**% Kill**	**Gooseberry Treated**	**% Kill**
Neat	* Not countable	90	-	270	-	280	-
10^−1^	12,700	200	98.42	300	97.63	300	97.63
10^−2^	20,000	1000	95	1000	95	1000	95
10^−3^	90,000	0	100	0	100	10,000	88.88
10^−4^	0	0	100	0	100	0	100
Panel 2							
**Dilutions**	**24 h**						
	**Control**	**Garlic Treated**	**% Kill**	**Clove Treated**	**% Kill**	**Gooseberry Treated**	**% Kill**
Neat	* Not countable	98	-	* Not countable	-	* Not countable	-
10^−1^	15,700	50	99.68	700	95.54	690	95.60
10^−2^	110,000	100	99.90	2900	97.36	4600	95.81
10^−3^	700,000	0	100	5000	99.28	5000	99.28
10^−4^	0	0	100	0	100	0	100

* Colonies are too small and too many to be counted.

**Table 3 gels-11-00023-t003:** A comparative study of the current work with published literature.

Published Literature	Current Study	Reference
Antifungal effect of black pepper, bay leaf, cinnamon, and cumin against *C. albicans.* Solvent extraction Application: Treatment of oral Candidiasis No ALT	Antifungal effect of garlic, gooseberry, and clove against *C. albicans* Aqueous extraction Application: Control of *C. albicans* growth in the Foley’s catheter ALT setup	[65]
Extracts: *Carum carvi*, *Coriandrum sativum*, *Anethum graveolens*, *Cinnamomum zeylanicum* barks, *Laurus nobilis* leaves, *Eugenia caryophyllata* flower buds Organism: Isolated from soil sample, *Aspergillus*, *Mucor*, *Penicillium* Application: Preservation of food No ALT	Extracts: Garlic, gooseberry and clove Organism: Clinical isolates of *C. albicans* Application: Control of *C. albicans* growth in the Foley’s catheter ALT setup	[66]
Extracts: Basil, garlic, cinnamon, rosemary, etc. Extraction: Ethanol and water Organism: *Fusarium solani*, *Rhizoctonia solani*, *Alternaria solani*, *Macrophomina phaseolina* Application: Damping off disease control in soil No ALT	Extracts: Garlic, gooseberry and clove Extraction: Water Organism: Clinical isolates of *C. albicans* Application: Control of *C. albicans* growth in the Foley’s catheter ALT setup	[67]
Extracts: Chilli, coriander, pepper, cumin and asafoetida Extraction: Ethanol Organisms: *Rhizopus azygosporus*, *Mucor dimorphosphorous*, *Penicillium commune*, *Fusarium solani* Application: Food No ALT	Extracts: Garlic, gooseberry and clove Extraction: Water Organism: Clinical isolates of *C. albicans* Application: Clinical ALT setup	[68]
A*f*LT (ALT) review article Focused on conventional antifungal drugs for *C. albicans* species	ALT research article Focused on control of *C. albicans* by natural antifungal agents	[69]
Liposomal amphotericin B-based A*f*LT (ALT) Double-lumen catheter for dialysis Directly tested on patient	Natural agents based on ALT Foley’s catheter for UTI In vitro studies	[70]
Central venous catheter Conventional antifungal drug In vivo ALT study in humans	Foley’s catheter Natural antifungal agents In vitro ALT study	[24]
Nano biofilms of *C. albicans* grown on a high-density microarray platform Cells within the on-chip biofilms had morphology and resistance to antifungal drugs similar to the cells grown on a 96-well plate Miniaturization, automation, minimum use of reagents, and reduced assay cost No ALT	*Candida* biofilms grown in Foley’s catheter *Candida* biofilms are resistant to conventional antifungal drugs In vitro set up with conventional/standard methods of analysis ALT setup	[71,72]
Development of an immune-based microfluidic device to detect *C. albicans* from PBS or human whole blood Rapid and inexpensive method 61–78% efficiency in the detection of *C. albicans* No ALT	In vitro Foley’s catheter-based model for growth and control of *C. albicans* biofilm Conventional method The % kill of *C. albicans* M207 with garlic—82.85; and *C. albicans* S470: Garlic—96.81, gooseberry—56.05, clove—55.41 ALT set up	[73]

## Data Availability

The original contributions presented in this study are included in the article/Appendix A. Further inquiries can be directed to the corresponding author.

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
