# Peer review of "In Situ Aqueous Spice Extract-Based Antifungal Lock Strategy for Salvage of Foley’s Catheter Biofouled with Candida albicans Biofilm Gel"

_gels, 2025, doi:10.3390/gels11010023_

Round 1
Reviewer 1 Report
Comments and Suggestions for Authors
Dear Authors,
Thank you for the interesting manuscript on the actual topic of antifungal protection. This area has been developed rapidly last few decades, nevertheless it still contains some urgent tasks, including searching novel active drugs. Therefore, I believe the manuscript can be interesting for the readers of the Journal in case of its revision. I have some comments on it:
1. The Authors claim the study is one of the first using spice extracts, while there are few papers already published on the topic of antifungal effect of spice extracts since 1990s:
https://journals.lww.com/ijcm/fulltext/2019/44001/antifungal_efficacy_of_spice_extracts_against.21.aspx;
http://fens.usv.ro/index.php/FENS/article/view/571;
https://www.plantprotection.pl/Antifungal-effect-of-powdered-spices-and-their-extracts-on-growth-and-activity-of,91115,0,2.html;
https://www.sciencedirect.com/science/article/pii/S0023643896900420
Also, the Authors do not refer to the highly-cited works considering ALT, e.g. https://journals.asm.org/doi/10.1128/aac.01351-12.
2. The Authors do not consider exploiting novel lab-on-a-chip techniques for C. Albicans drug resistance studies, while this approach is being actively developed worldwide, e.g.
https://www.nature.com/articles/s41598-019-44298-w;
https://pubs.rsc.org/en/content/articlelanding/2022/lc/d2lc00219a;
https://www.sciencedirect.com/science/article/pii/S0142961219304958;
https://www.mdpi.com/1424-8220/19/15/3366, etc.
Therefore the overall novelty of the current study, especially in part of experimental techniques, seems to be insufficient for the Journal. I recommend to stress additionally the novelty of the study.
I recommend to add a comparison to the other results, including those considering biofilms studies, e.g.
https://journals.plos.org/plosone/article?id=10.1371/journal.pone.0019036;
https://app.jove.com/t/3845/candida-albicans-biofilm-chip-cabchip-for-high-throughput-antifungal;
https://pubs.acs.org/doi/10.1021/acsomega.9b00499.
3. The goal of the current study is recommended to be given in more compact form in the Introduction section if order to estimate its success in the Conclusion section.
4. The catheter photo in Figure1, in my opinion, does not provide any essential information and can be omitted.
5. To show the original results of the current study, a comparative table with the literature and the own results has to be introduced to the Discussion section.
6. The font size in the scale bars in Figure 5 panels is too small and needs to be enlarged. Otherwise, SEM images do not provide any quantitative information.
7. The text font size in Figure 6 is also too small and needs to be enlarged.
8. Since the Authors use standard test techniques to estimate the efficacy of antifungal therapy, I recommend describing in more detail the used original or commercial software for CFU analysis and the other tests.
9. The manuscript contains typos (e.g. "a self-produced gel matrix", Line 43; Ref. 6 in the References section, Lines 480-481) and needs to be thoroughly revised before resubmitting.
Reviewer 2 Report
Comments and Suggestions for Authors
Abstract
- Take into consideration to reorganize it including relevant sections: Background, Objectives, Material and Methods, Results, Conclusions to be easier to follow it.
Introduction
- It will be useful to add a list of abbreviations
- Please place in a broader context the importance of Hospital Acquired Infections highlighting some epidemiological data
- Please mention the name of the most frequent bacterial/ fungal pathogens involved in CAUTI
- Some phrases should be reformulated to make more sense (for example lines 44 - 46)
- The aim of the study should be better sustained and highlighted
- The paragraph between lines 87-108 should be focused more on the antimicrobial potential of the extracts than on the enumeration of their chemical composition
Material and Methods
- This section is well organised, the methods are described clearly and in sufficient detail.
Discussion
- This section describes in detail the proposed mechanisms for the antifungal activity of the extracts, aspect very important in this context. However, I suggest to discuss more deeply the obtained results and to compare with existing publications, if available.
Round 2
Reviewer 1 Report
Comments and Suggestions for Authors
Dear Authors,
Thank you for addressing my comments.
I think now the manuscript is ready for publication.
Reviewer 2 Report
Comments and Suggestions for Authors
The manuscript was revised according to the suggestions.